# The Use of the Internet of Things for Estimating Personal Pollution Exposure

**DOI:** 10.3390/ijerph16173130

**Published:** 2019-08-28

**Authors:** Keith April G. Arano, Shengjing Sun, Joaquin Ordieres-Mere, and Bing Gong

**Affiliations:** 1Department of Management, Economics and Industrial Engineering, Politecnico di Milano, Via Lambruschini 4/B - (Building 26/B), 20156 Milan, Italy; 2Department of Industrial Engineering, Business Administration and Statistics, E.T.S de Ingenieros Industriales, Universidad Politécnica de Madrid, Calle de Jose Gutierrez Abascal 2, 28006 Madrid, Spain; 3Jülich Supercomputing Center Forschungszentrum Jülich GmbH, Wilhelm-Wohnen-Str, 52425 Jülich, Germany

**Keywords:** Personal Air Pollution Exposure (PAPE), air pollution monitoring, IoT, Air Quality Decision Support System, health impact

## Abstract

This paper proposes a framework for an Air Quality Decision Support System (AQDSS), and as a proof of concept, develops an Internet of Things (IoT) application based on this framework. This application was assessed by means of a case study in the City of Madrid. We employed different sensors and combined outdoor and indoor data with spatiotemporal activity patterns to estimate the Personal Air Pollution Exposure (PAPE) of an individual. This pilot case study presents evidence that PAPE can be estimated by employing indoor air quality monitors and e-beacon technology that have not previously been used in similar studies and have the advantages of being low-cost and unobtrusive to the individual. In future work, our IoT application can be extended to include prediction models, enabling dynamic feedback about PAPE risks. Furthermore, PAPE data from this type of application could be useful for air quality policy development as well as in epidemiological studies that explore the effects of air pollution on certain diseases.

## 1. Introduction

Pollution and various forms of ecosystem contamination continue to be pressing issues across the globe [1]. China’s rapid increase in urbanization in the last three decades, for example, has resulted into environmental challenges where air pollution is the leading problem [2]. Protecting the environment, therefore, is a serious undertaking that faces businesses and governments today. In recent years, there has been increasing pressure on institutions to measure and report environment-related parameters [3]. For this reason, there has been a significant increase in the number of reporting instruments used globally, of which sustainability reporting instruments account for the largest share owing to government regulations [4].

Environmental sustainability now underpins the policy-building initiatives of government institutions and businesses alike. In developed countries such as those of the European Union (EU), air pollution damage, which brings about a direct threat to public health, is expected to rise in the next decade. This has compelled the EU governments to give priority to air pollution level reduction above any other climate change policy plans [5]. In developing countries, however, there are still inadequate air quality policies and environmental monitoring plans. This is a major concern primarily because these are the regions that are more susceptible to increasing levels of air pollution [6]. There is therefore a challenge in finding economical solutions to monitor pollution levels and other relevant health parameters.

Air pollution from both outdoor and indoor sources constitutes the greatest environmental risk to human health around the globe [7]. About seven million premature deaths were attributed to air pollution in 2014, based on an estimate by the World Health Organization (WHO). It is projected that, by 2050, outdoor air pollution will be the number one cause of environment-related deaths worldwide [8]. For these reasons, governments across the globe have started to monitor the levels of major air pollutants, especially in metropolitan and urban areas.

In addition, people spend around 90% of their time indoors, and human exposure to indoor air pollutants may occasionally be more than 100 times higher than outdoor pollutant levels, according to the United States (US) Environmental Protection Agency (EPA). Indoor air pollution is equally detrimental, as statistics show that 4.3 million people per year die from exposure to household air pollution [9]. Exposure to poor indoor air is a significant cause of productivity loss in the US, as productivity decreases by 0.5 to 5% per workplace, generating a loss of 20 to 200 billion US dollars annually [10].

Monitoring Personal Air Pollution Exposure (PAPE), which refers to the amount (μg) of pollution being inhaled by an individual, has been a topic of growing interest worldwide, not only as a result of global health policies, but more importantly, due to the interest in understanding its effects on various cardiovascular and respiratory diseases. These diseases have been documented widely in existing epidemiological studies [11,12]. Nevertheless, this traditional evaluation of PAPE has not been directly undertaken for individuals, but rather for groups of people exposed to the annual average concentrations of pollution that are indicated by a network of fixed-site outdoor monitors.

Existing studies on the association of air pollution with different diseases [13,14] recognize the importance of measuring PAPE among individuals. By monitoring activity patterns, it is possible to establish correlations between different populations or levels of socioeconomic status and PAPE. Although there are novel methods to measure and model these exposures, the great variability in PAPE remains a major challenge [15] and provides a compelling case for research on the effects of air pollution on health. Dias and Tchepel [16] suggest that, in order to assess personal exposure, not only the spatial-temporal variability of urban air pollution should be taken into account but also the indoor exposure and the individual time-dependent activities should be measured.

The recent advances in data technology are expected to play major roles in the next decade, permitting easier access and analysis of data [1]. The Internet of Things (IoT), which is defined as the network of various ubiquitous devices that are capable of computation and communication over the Internet, has been gaining recognition in the development of advanced applications in the healthcare sector [17]. This wave of digital innovation is driving the healthcare industry and paving the way for cheaper, smaller and more efficient wearable technologies that monitor health indicators in real time [18].

Health systems around the world are under pressure to come up with economical solutions to existing problems [18]. These sensor technologies, coupled with sophisticated analytics, have the means to improve process efficiency and achieve cost reductions [19]. With this kind of digital innovation, an economical and scalable application can be built to help developing countries to measure PAPE and other health parameters to improve their current health risk assessment systems.

Although there was initially a concern about sharing private information from sensor technologies or other similar devices, the general public has started to accept the use of digital services, even for sensitive information such as health data [20]. In fact, in a consumer survey performed by Price Waterhouse Coopers in 2015, 83% of respondents indicated that they were willing to share data to aid in the diagnosis and treatment of diseases [21]. With this collaborative support from the general public, exploring IoT applications to test sensor-driven projects can greatly facilitate advances in the healthcare industry [22]. Insights gained from the surveillance of vital health information, such as PAPE, can establish a foundation for predictive, preventive, and personalized healthcare systems [18].

In line with these trends in the healthcare industry, this paper seeks to propose a framework for an Air Quality Decision Support System (AQDSS) and to develop an IoT application that measures PAPE based on this framework. The four sections of this paper are organized as follows. The first section provides a literature review of studies on key air pollutants, determination methods for air pollutants, and PAPE estimation techniques as well as the current opportunities and challenges in the field. The second part discusses the methodology, which includes the proposed framework and an IoT application that was tested by means of a case study. This is followed in the third part by the analysis and a discussion of the results of the case study. The fourth part presents conclusions that highlight the study’s important contributions and directions for future research.

## 2. Literature Review

### 2.1. Key Air Pollutants

The key air pollutants that are currently being monitored by agencies such as the WHO, the EPA in the United States, and the European Environment Agency (EEA) in Europe, are particulate matter (PM${}_{10}$ and PM${}_{2.5}$), ozone (O${}_{3}$), nitrogen oxide (NO), nitrogen dioxide (NO${}_{2}$), carbon monoxide (CO), sulfur dioxide (SO${}_{2}$), volatile organic compounds (VOC), and benzene (C${}_{6}$H${}_{6}$). They are also frequently studied in academic research [23,24]. Although there is a substantial amount of monitoring data available for each of these pollutants. PM${}_{10}$ and PM${}_{2.5}$ are considered to be the most widely studied air pollutants in the existing environmental risk and health literature. This is because PM poses one of the greatest risks to human health [25].

The indoor environment is a critical domain where an average person spends an estimated 90% of his or her time [26]. Thus, indoor air pollution is more likely to account for total population exposure than pollution from the outdoor environment [27]. While individuals are spending more and more time indoors, an assessment of the health impact of indoor air pollution has not been studied as extensively as the impact of outdoor air pollutants. One of the main reasons for this is the lack of indoor air quality monitoring information [28]. There are primary indoor air pollutants, which are recommended based on the EU (2008) directive for Clean Air and the WHO [28]. They are schematically listed as benzene, formaldehyde, naphthalene, nitrogen dioxide, polycyclic aromatic hydrocarbon, radon, trichloroethylene, and tetrachloroethylene. In the research community of indoor air quality monitoring and assessment, particulate matter, carbon dioxide, carbon monoxide, ozone, nitrogen oxide, formaldehyde, benzene, total volatile organic compound (TVOC), polycyclic aromatic hydrocarbon, and other VOCs have been extensively studied [29,30].

### 2.2. Determination Method for Air Pollutants

Most workspaces or industrial environments still apply traditional measuring strategies to assess occupational health and safety. These strategies are mainly based on the EPA Compendium of Methods [31] and the International Organization for Standardization (ISO) method, which rely on complex sampling and analysis techniques. These methods, such as Method-10A and IP-3A, require domain experts to prepare diffusive or passive samplers and are frequently replaced with new ones due to the limited equipment lifespan. Subsequently, the collected samplers are separated by gas chromatography and measured by mass-selective detector or multidetector techniques in a remote laboratory [32]. Moreover, to measure multiple pollutants, the equipment for each pollutant has to be prepared or bought from different manufacturers, which can lead to issues about data manipulation and integration. These aforementioned elements of the traditional measuring strategy restrict the sampling time to a short-term basis [33]. As indoor air quality varies from time to time due to changes in working conditions, human activity, and weather conditions, short term sampling cannot cover all kinds of variations. Therefore, long-term monitoring has become a need in the research community and practical applications such as Occupational Safety and Health (OSH) management.

The rapid development of IoT and sensor techniques enables light, low-cost, and real-time pollution monitoring solutions. The integration of IoT and the sensor network in air quality monitoring addresses the aforementioned gaps: short-term monitoring and complex air monitoring solutions. Recent studies on the development of indoor air quality monitoring systems have been undertaken on PM, carbon dioxide (CO2), CO, and VOC. Moreover, IoT-based indoor air monitoring devices such as Foobot and AirVisual are already commercially available on the market.

### 2.3. IPAPE Measurement Techniques

As noted previously, there is a growing interest in measuring PAPE at the individual level. At present, there is a wide range of low-cost sensor technologies [34] that can be leveraged to implement large scale monitoring networks by means of complex measurement techniques [35]. PAPE requires tracking of a person’s activity patterns to learn the time and location of their exposure to pollution concentrations as well as the duration of exposure and nature of the pollutants. This is necessary to understand the probable effects on health of the exposure [36].

The different PAPE measurement techniques that have been developed in the last decade can be grouped into three categories. The first group is the traditional method in which pollution data are collected from fixed-site outdoor monitors and assigned to the home address of the individual through spatial interpolation techniques. Examples include Land Use Regression (LUR) [37], Inverse Distance Interpolation [38], and the geostatistical Kriging algorithm [39]. Numerical models, such as the Community Multiscale Air Quality (CMAQ) model and the Urban Atmospheric Dispersion model (DAUMOD), were proposed for regional air pollution modeling prediction in previous studies [40]. However, the expensive computational cost and failure to capture pollution variability make them inadequate for the application of modeling in real time in urban areas where there are severe photochemical pollution conditions. Graz Lagrangian Model (GRAL) is another advanced mathematical model that can handle the motion of pollution in buildings and complex terrains [41]. A major drawback of these types of models, however, is the need to have accurate information about emissions, meteorological data, and the structural and geographical figures of the area, which may not always be available in high resolution [42]. While the performance of the spatial interpolation methods may significantly drop in dynamic terrains such as in urban environments, they have still been used widely in recent studies [43,44] of areas where the detailed information needed for complex numerical models (e.g., street-based monitoring) is still unavailable.

In summary, these methods are inadequate, as they do not address the issue of the individual’s spatio-temporal PAPE variability [45] and neglect indoor air pollution. Accordingly, this has led researchers to explore new techniques that can provide more accurate measures of PAPE.

The second group of techniques, which is built on the traditional method but addresses the issue of exposure variability, takes into account the activity patterns by tracking an individual’s location. It incorporates indoor pollution data based on the amount of time spent indoors. A commonly used indoor pollution measurement method is the indoor/outdoor ratio [37]. Other techniques, such as modeling based on data from vehicle type and emissions have also been proposed [46]. With respect to activity tracking, different tools have been used in studies to track the location and activity patterns of an individual. These include Global Positioning Systems (GPS) [35], public WiFi networks [37], and accelerometers [45]. A common characteristic shared by these activity tracking tools is the use of a mobile device, particularly a smartphone. This mobile technology has proved to be an enabling tool in the health industry with its ability to access data anytime from anywhere [47]. Although this group of PAPE measurement techniques is an improvement from the first group, it still faces the issue of pollution variability and measurement accuracy with its reliance on fixed-site outdoor monitors and indoor/outdoor ratios alone [48].

The last group of techniques stems from the two previously discussed groups but further captures the issue of indoor pollution measurement accuracy and the spatio-temporal resolutions of data from fixed-site outdoor monitors. The periodic measurements by these fixed-site outdoor monitors by nature have low spatial resolution and do not address the issue of variability in pollution concentration [49]. Since the indoor environment has a much greater impact on human health than the outdoor environment [50], it is essential to have solutions that can provide more accurate measures of indoor air pollution instead of employing the traditional method of using the indoor/outdoor ratio.

Personal exposure measurements can be performed directly and indirectly [51]. Passive samplers are widely used in personal sampling, since they have the merits of being light, electricity-free, and wearable. Passive samplers exist for nitrogen dioxide, carbon monoxide, VOC, ozone, sulfur dioxide, and formaldehyde [52]. Due to sampler lifespan, the sampling time usually lasts from a few days up to one week [53].

On the other hand, using a micro-environmental model is an indirect way of assessing personal exposure. In daily life, people move around and are exposed to various levels of pollutants in various locations. The term “micro-environment” is defined as a chunk of air space with a homogeneous pollutant concentration [54]. Such a micro-environment can be an indoor location (bedroom, kitchen, etc.) or workplace location (meeting room, office, printing room, etc.). The spatio-temporal individual time-activity crossing in micro-environments is tracked through questionnaires or time-activity diaries (TADs).

The key to measuring individual pollution exposure is to track an individual’s activities in both the space and time dimensions. GPS technology is the ideal technology, and it has been used successfully for this purpose. Some well-designed integrations of GPS devices and portable pollution monitors have been proposed by some studies [35,55] to determine the potential exposure at the individual level. However, in indoor environments, GPS technology does not function as well as it does outdoors.

Therefore, more extensive approaches have been developed such as the use of mobile sensors (i.e., handheld, USB-pluggable smartphone sensors, wearable sensors) to monitor PAPE indoors. Studies [56,57] that have included this group of PAPE measurement techniques have managed to address the most relevant issues of pollution variability by employing mobile sensors. Beacon technology offers a promising solution for indoor location tracking. Furthermore, the use of indoor monitors instead of mobile sensors, which are often used in similar studies, eliminates the inconvenience of carrying a device around. The use of an indoor monitor and e-beacons also enables unobtrusive and low-cost collection of pollution data for multiple individuals, in contrast to a mobile sensor, which only collects data for a single individual.

### 2.4. Opportunities and Challenges

Although continuous technological advancements have enabled researchers to propose solutions that provide measures of PAPE, the issue about the cost and scalability of such methods remains to be addressed. The most recent approach, as discussed in the third group of PAPE measurement techniques, employs mobile sensors that the individual carries around. Although these mobile sensors are able to provide better spatial resolution of pollution data, the willingness of individuals to carry these sensors is still a challenge, in addition to the cost and scalability issues of the method.

The trade-off between cost and quality of pollution data continues to be a point of discussion among studies. The proposed PAPE measurement techniques that are currently available in the literature are limited to PAPE estimation alone and, therefore, fail to provide a more comprehensive view of the entire AQDSS. Thus, there is an opportunity to further explore the use of existing technologies to enable the development of a more comprehensive PAPE measurement technique that is able to provide a preventive, predictive, and personalized system.

Within similar studies on the measurement of PAPE, there are some proposed conceptual frameworks [35,36] and system architectures [56,58]. However, they are centered primarily on PAPE measurement and the potential health impacts. In this paper, we present a comprehensive framework that encompasses not only PAPE measurement but provides a holistic view of the entire AQDSS. As a proof of concept for this framework, we also develop a low-cost and unobtrusive IoT application for measuring PAPE that addresses the gaps in the currently available solutions.

## 3. Methodology

### 3.1. Framework

Figure 1 shows the proposed framework for an AQDSS. There are three key stakeholders identified, namely the individual, the healthcare industry, and the government. The three pillars at the center represent the elements that are directly linked to the government. They include pollution laws, sectorial regulations, and incentives, all of which make up the air quality monitoring policies of the government.

As pointed out previously, these government regulations play an important role, as they largely support and drive policies that enable the measurement and access to air quality data.

The other two stakeholders are the individuals and the healthcare sector, which form the apex of the framework. They are supported by five layers of activities, as illustrated by the pillars on both sides. The first three on the left correspond to the analysis of past data to estimate PAPE and their related health impacts. The two pillars on the right represent future possibilities of forecasting PAPE and the associated health risks.

Although the estimation of PAPE is fundamental to the entire AQDSS, its associations with forecasting and as a predictive health risk assessment system are noteworthy. Air quality forecasting techniques are already being explored in current research [59,60] in environmental modeling literature, where their important contributions to the development of control measures to prevent damage to human health have been highlighted.

This proposed framework can be adopted to aid in the development of an AQDSS and various IoT applications. For instance, consider a mobile application that allows an individual to select the best route to travel from home to work that minimizes the risk to pollution exposure. This could be achieved by employing different modeling techniques to continuously analyze real-time air quality data and forecast PAPE values for each of the possible routes to the destination. Actual PAPE data that are stored in the database can also be used for epidemiological studies and air quality policy development. This could be one of the applications of the proposed framework when all identified pillars are fully employed. In this paper, however, as a proof of concept, we focus mainly on PAPE measurement (pillar 2) in accordance with the system architecture that is illustrated on the base of the framework. In order to manage sensor data in an interoperable way, this implementation considers the Web Service Description Language provided by the Sensor Observation Service v2.1 (SOS) from the SOS-OGC consortium. This standard defines a Web service interface which allows observation queries, sensor metadata, as well as representations of observed features. Furthermore, this standard defines a means to register new sensors and to remove existing ones. Also, it defines operations to insert new sensor observations. The feasibility was assessed through the developed case study.

#### 3.1.1. System Architecture

As indicated in the framework, the base shows the set of activities that are related to the gathering and management of all air quality and personal data. This groundwork is required for the entire system to function. There are five different sources of data. They are the (S1) outdoor pollution monitors, (S2) location tracking application, (S3) indoor pollution monitors, (S4) e-beacons, and (S5) meteorological monitors. S1 and S2 are intended for outdoor pollution modeling, and S3 and S4 are for indoor pollution modeling. We also consider meteorological data, as they are relevant for air quality prediction studies [60,61].

The data are extracted from the mentioned data sources and stored in a database management system. Data mining, numerical modeling, and geostatistics, as shown in the center of the framework are the key activities that support the entire system, as it is a continuous process to discover and analyze spatio-temporal data.

#### 3.1.2. PAPE Measurement

There are essentially three elements to consider when measuring PAPE. They are (1) outdoor pollution, (2) indoor pollution, and (3) the individual’s location pattern. With respect to the outdoor pollution, the mobile-phone-based tracking app provides the time and location data of the individual in the outdoor environment. An outdoor pollution map is created by using potentially different strategies, such as

Numerical-modeling-based dispersion models [62,63],Big data, machine-learning-based models [60,64],Geostatistic-based techniques, like Kriging [65,66].

Each of these techniques has its specific advantages and limitations, and its consideration in a specific application will support its choice. Actually, the latest family of methods has specific advantages, as it is suitable for working with the fixed network of pollution stations the city has implemented. Indeed, it also deals with the limitations of sparse data, as Data fusion can increase the reliability of data as well as it can contribute to dealing with local effects like street canyons, etc., by using the street granularity-based IoT air quality stations some cities are deploying, such as Airbox in Taipei [67] and Array of Things (AoT) sensor boxes in Chicago [68].

For the duration of time that an individual is outdoors, the corresponding pollution data are estimated by superimposing the developed outdoor pollution map over the collected location pattern data.

For the indoor pollution, the e-beacons indicate the period when the individual is indoors, and the indoor air quality monitors provide the corresponding air quality data, when available. Failing this, outdoor information will be used by default. The integration of personal mobiles and fixed e-beacons located in different indoor micro-environments enables the individual’s time-location information to be understood. The corresponding time-location knowledge combined with location-specific indoor air quality information collected from air monitoring devices can provide a detailed picture of personal exposure in the indoor environment.

Both outdoor and indoor data are then integrated, and statistical modeling techniques are employed to either estimate or forecast the individual’s PAPE.

### 3.2. Madrid Case Study

In order to assess the feasibility of the proposed IoT application that measures PAPE and contributes to empowering users because of the relevant figures provided at the personal level, we conducted a case study to analyze significant functionalities.

#### 3.2.1. Study Area

The study area was the City of Madrid, which is the capital of Spain, as well as its largest municipality. It was the first city in Spain to have air quality monitoring stations and has always been at the forefront of the fight against air pollution. In response to the most recent EU directive (Directive 2008/50/EC) regarding the establishment of limits to major air pollutants, the Madrid government has committed to maintaining acceptable pollution levels by continuous air quality monitoring.

The Madrid air pollution monitoring network consists of 24 fixed-site outdoor monitors (Figure 2). The hourly averaged measurements of SO${}_{2}$, CO, NO, NO${}_{2}$, PM${}_{10}$, PM${}_{2.5}$, C${}_{6}$H${}_{6}$, toluene (C${}_{6}$H${}_{5}$–CH${}_{3}$), hexane (C${}_{6}$H${}_{14}$), propene (C${}_{3}$H${}_{6}$), m-xylene, o-xylene, and methane (CH${}_{4}$) hydrocarbons can be downloaded free of charge from the official open data website of the *Ayuntamiento de Madrid* [69]. Meteorological data, such as temperature, humidity, ultraviolet radiation, pressure, solar radiation, rainfall, precipitation, diffuse solar radiation, global radiation, wind speed, and wind direction can also be accessed through the website of the *Agencia Estatal de Meteorologia* [70].

#### 3.2.2. Data Collection

In this case study, S1, S2, S3, and S4 data sources were used and meteorological data were excluded (see Figure 1). We had one individual volunteer whose activities were monitored during the study period.

For this particular case, Madrid does not yet implement the street-based pollution monitoring strategy, but based on similar studies [43,44], the research team adopted the geostatistics-based approach, as it becomes linear scalable with time and is suitable for integrating additional data sources. Therefore, outdoor pollution figures were downloaded from the mentioned open data website of Madrid City Hall. For location tracking, we used the mobile app Moves [71] in which time, location, and activity were accessed through an open Application Programming Interface (API). Other similar open source mobile apps are widely available, such as OwnTracks [72], Miataru [73], and Geo2Tag [74].

For the indoor pollution, Foobot indoor monitors were used. One of them was placed in the individual’s workplace, as this is where she spends most of her indoor time. The indoor pollution data were retrieved from Foobot’s API [75]. The e-beacon devices were placed in proximity to the indoor monitors. They helped us to determine whether the individual was within the indoor vicinity. The e-beacon data were broadcasted through Eddystone, an open-source beacon format, and were retrieved through an app that we developed in Cordova [76]—a free and open-source platform for building mobile applications.

All of these data sources promote the scalability of the proposed IoT application, as most are publicly available without charge. The only costs incurred were for the indoor monitor and e-beacons. E-beacons, however, are low-cost, small enough to attach to any surface, and are finding an increasing number of location-based applications in various industries such as retail and transportation as well as in households [77]. Hence, beacon technology offers a promising solution for indoor location tracking.

All data that were collected from the mentioned data sources were processed as indicated in the Data in Brief Collection documents that were submitted to the journal for this paper. The developed code can be also found in a public repository [78].

The selection of the pollutants used for the PAPE estimation was based primarily on the data provided by the devices, which also agreed with the data on the most common air pollutants that have been widely studied previously [23,24]. Table 1 shows the available pollutants for each of the data sources used.

#### 3.2.3. Outdoor Pollution Modeling

Existing studies of PAPE essentially rely on modeling techniques in which data collected from fixed-site outdoor monitors are used to estimate pollution at specific geographic locations. To create an outdoor pollution map, there are several alternative methods. These include using micro meteorological numerical-based models (WRF, CMAQ, etc.) [79], or machine-learning-based models [64]. However, for the sake of simplicity and considering the computational costs and the number of potential users, we adopted some classical but still cost-effective approaches like the Inverse Distance Weighting (IDW), Simple Kriging, Ordinary Kriging, and Co-Kriging algorithms.

Table 2 shows the formulae and main characteristics of these techniques in which $\hat{z_{0}}$ is the measured value at the prediction location, $\lambda_{i}$ is the weight of the measured value at the ith location, and $X_{i}$ is the measured value at the ith location. The parameters that were tuned are also indicated in the table.

All three methods estimate the value at a particular location by assigning a weight of the surrounding known values and calculating the weighted sum of the data. These techniques differ mainly in the calculation of the assigned weight $\lambda_{i}$. Kriging, which is a geostatistical method, offers advantages over other interpolation techniques, as it provides an interpolation error estimate, and it is an exact interpolation. The interpolations are based on weights that do not depend on data values [80]. The advantages of the deterministic interpolation technique IDW, on the other hand, are that it is simple, intuitive, and computes the interpolated values quickly [81]. We created the outdoor pollution map by employing these three interpolation techniques in R, an open-source statistical modeling software.
Optimal Parameters and Model SelectionIn order to select the optimal parameters and the best modeling technique for each of the hourly outdoor pollution datasets, a 5-fold cross validation was performed to avoid overfitting. For each of the 24-hourly datasets and each of the three modeling techniques and all combinations of their respective parameters, the selection of optimal values was based on the root-mean-squared-error (RMSE) metric. The dataset was separated into two parts, training and testing, which were used to fit the model and calculate errors, respectively. The parameters and the model that provided the least RMSE were selected.For the Simple and Ordinary Kriging techniques, the weights $\lambda_{i}$ were derived by fitting a covariance function or variogram. First, a graph of the empirical variogram was plotted and a model was fitted to the points based on this plot. Table 3 shows the different models and functions from which to choose when fitting a model to the empirical variogram. Based on the 5-fold cross validation, the Gaussian Model was selected as the optimal configuration.Outdoor Pollution MapSimilar to [43], an hourly outdoor pollution map was created that was based on the identified optimal parameters and modeling technique for each respective hour. Figure 3 shows an example of the pollution maps based on the PM${}_{2.5}$ pollution data on 2017-03-24. It shows that, from midnight to the morning at around 6:00, the highest pollution levels consistently occurred in the southwestern part of the city and moved towards the north with maximum levels that ranged from 8 to 12 μg/m${}^{3}$. Concurrently, high pollution levels were also experienced in the northwestern part of the city at midnight and in the northeastern part at 01:00 in the morning.The selection of time frequency (hourly-based in this case) also impacts the accuracy, depending on how spiky the pollution looks. In Madrid, the pollution sources are strongly related to traffic and then variations are smooth [82]. Therefore, hourly-based frequency is a rather convenient basis for calculations.


#### 3.2.4. Indoor Pollution Modeling

The main data sources used to model the indoor pollution were the e-beacons and indoor monitor. The timestamps recorded from the e-beacons provide the time when the individual was detected indoors.

In this study, we refer to “indoor” as the work location, since the indoor monitor was only present at the individual’s workplace. The “outdoor” environment, on the other hand, refers to any other location outside the workplace. To obtain the corresponding pollution values during these periods, each of these timestamps was matched to the closest timestamp logged from the Foobot device. As illustrated in Figure 4, the pollution values were then aggregated in time periods based on Equation (Equation 1) on the assumption that, if the difference between two sequential timestamps recorded on the e-beacons was more than 10 min, the individual was outdoors and a new indoor period would start. For instance, in Figure 4, from 2017-03-24 at 14:37:46 to 2017-03-24 at 14:44:59, the pollution levels were aggregated, since the timestamp immediately following 2017-03-24 at 14:44:59 is 2017-03-24 at 14:59:00 and the difference is longer than 10 min. Therefore, during the period between 2017-03-24 at 14:44:59 and 2017-03-24 at 14:59:00, the individual was outdoors and the new indoor period resumed at 2017-03-24 14:59:00.

Similarly, micro-environments (office, printing room, meeting room) in the workplace could be replicated by deploying e-beacons and air monitoring devices in all available micro-environments.

The PAPE Exposure(p) in period p inhaled by the individual was calculated by multiplying the pollution value SZ(p) by the respective minute ventilation (VE) value using Equation (Equation 2).
(1)$$SZ\left(p\right)=\sum_{t=i}^{n}\frac{1}{2}\left(Z_{t_{i+1}}\right)+Z_{t_{i}})\left(t_{i+1}-t_{i}\right)$$
where $t_{i+1}-t_{i}<10$ mins, and SZ(p) is the fully aggregated pollution value during the period from time $t_{i}$ to $t_{n}$. This period is named *p*.
(2)Exposure(p)=SZ(p)*VE
where VE∈(VER,VEW,VERT,VEC)

VER = VE for activity type “run”;

VEW = VE for activity type “walk”;

VERT = VE for activity types “rest” and “transport”;

VEC = VE for activity type “cycle”.

#### 3.2.5. Indoor and Outdoor Pollution Integration

The individual’s location was tracked through the Moves mobile application. The recorded data from this tracking app include the starting and ending times, latitude, longitude, and activity type, as shown in Table 4. To obtain the corresponding pollution values for these periods, the time and location records were matched against the interpolated values from the created outdoor pollution map. The resulting outdoor pollution data were then matched against the aggregated indoor pollution values in Figure 4, in which the outdoor data were replaced by the corresponding indoor data.

Table 5 shows the resulting individual’s indoor and outdoor PAPE values for PM${}_{2.5}$ with the respective period (i.e., starting and ending times), location (i.e., longitude and latitude), environment type (i.e., indoor or outdoor), activity type (i.e., transport, rest, walk, run, cycle), and minute ventilation (VE). The PAPE values are indicated in its last column ”Exposure”.

VE (m${}^{3}$/min) measures the volume of gas inhaled by an individual. It varies with the type of activity. The type of activity or travel mode may have a significant effect on the exposure values [83,84] and, hence, it is important to account for VE. We obtained the VE values from a study done by [85] on human inhalation rates. The types of activities in the tracking app include “transport”, “walk”, “run”, and “cycle” and are based primarily on the speed of movement of the individual. In this study, for the time periods that lack one of these types of activity data, we assumed that the individual was at “rest” (i.e., sleeping, sitting, etc.). Since VE is based primarily on the body movement of the individual, we used the same VE values for both activity types “transport” and “rest”.

#### 3.2.6. Practical Application

To illustrate a possible IoT application [86,87] that can be developed using the proposed framework, we identified different travel routes and their corresponding forecasted PAPE values [60] that give the individual an opportunity to select a travel route that minimizes the risk of exposure to pollution.

As an example, we selected an entry in Table 5 for the time period 12:04:30 to 12:23:44 on 24 March 2017, in which the individual was outdoors and in transport mode. During this selected period, by using the starting and ending location data that the tracking app provided, we identified alternative routes using the ggmap package in R.

From this package, the estimated travel time and route locations (i.e., latitude and longitude) were obtained. Then, based on these specific time and location data, the corresponding pollution values were taken from the previously interpolated outdoor pollution values.

## 4. Results and Discussion

### 4.1. Outdoor Pollution Model Performance

The adopted modeling technique based on geostatistics [65,66] using hourly-based data [43] from the fixed network of pollution stations can be interpolated by using different techniques, and criteria for technique selection is needed. Therefore, a cross validation with the leave out strategy was adopted. Based on the 5-fold cross validation performed for each of the three modeling techniques, among the 24 hourly datasets of PM${}_{2.5}$ captured on 34 March 2017, the Simple Kriging technique proved to be the best model with a selection occurrence of 13, followed by the Ordinary Kriging with 10, Co-kriging with 7, and IDW with 1, out of 24 datasets. Our results agree with previous studies such as [88], where Simple Kriging outperformed Co-Kriging, and [89], in which Simple Kriging turned out to be the best model for estimating NO${}_{2}$ and PM${}_{10}$.

It can be argued that some local effects like turbulence around buildings, roughness of constructions, and some other aspects impact the accuracy of the estimation. The techniques used in such dimensions can be those related to the integration of meteorological, chemical and transportation numerical modeling (WRF and CMAQ models), with the limitations of being able to precisely estimate the boundary conditions as well as to properly model the city configuration (buildings, trees, surface properties, etc.). When running with high spatial resolution, they produce good results, although the quality is slightly reduced and numerical stability becomes an issue [90]. Another potential contribution could be to use artificial-intelligence-based models to estimate pollution levels. In these fields, the authors have already made significant contributions. Actually, some papers [91] have shown the competitive advantage of these methods over those based on numerical simulations. However, to keep the implementation interoperable and extendable, interpolation was finally adopted, because it can easily be enriched with the data fusion option based on IoT-based, street level pollution sensors.

### 4.2. Device Performance

To validate the fully aggregated indoor pollution values (SZ(p)) obtained from the indoor monitor and e-beacon devices, they were matched against the pollution data that were measured simultaneously during the study period using a portable air pollution monitoring tool- Atmotube [92] that was carried by the individual.

Figure 5 shows the indoor VOC values measured from the Atmotube and the Foobot monitor on 2017-03-31. It can be seen that there is significant measurement variance between the two devices. Nevertheless, the measured values follow the same trend. There is no consistent Air Quality Index (AQI) provided for comparing the pollution values measured by each device. In agreement with [93], the AQI scales differ across countries, organizations, and devices, and this presents an obstacle for comparison and invalidates its usability, which emphasizes the need for a standardized awareness procedure.

Variance in measurements can be attributed to the differences in calibration and measurement methods that were used by these sensors (see Figure 5). However, this situation partially unveils the observed difficulties in getting people aware of the real importance of pollution, as someone can exhibit different figures for the same pollutants at the same place and point in time. Actually, it is another strong point to have a common framework such as the one proposed in this paper, because it mainly fosters transparency and then allows interpolated or modeled values for outdoor pollution over time at a particular place to be compared with local, privately owned sensors from both outdoor, and indoor locations. From such observations where different local sensors can indeed participate, a better understanding about outliers and commonalities and trends can be derived.

### 4.3. PAPE Values

Figure 6 illustrates a color map of the average PM${}_{2.5}$ levels (μg/m${}^{3}$) for one day, in which the range of the specific values is presented on a color scale on the right. The location pins indicate the environment, activity type, time percentage (%), and the respective amount of PM${}_{2.5}$ (μg) that the individual was exposed to within the indicated time duration. It can be seen that the individual spent most of the day (62.78 %) outdoors (i.e., outside the workplace) on the northwest side of the city where the highest daily average pollution level of 12 μg/m${}^{3}$ was concentrated, and this resulted in a total PM${}_{2.5}$ exposure of 52.7 μg.

Figure 7 shows the one day PM${}_{2.5}$ exposure levels by activity type. Based on this plot, the individual spent most of the day (88.13%) at rest and was exposed to approximately 70 μg of PM${}_{2.5}$ during this period. PM${}_{2.5}$ exposure values within a selected time period on the same day are also plotted in Figure 8, which shows that the individual had the highest pollution exposure at 15:32 in the afternoon during this selected period.

From this analysis, the value of people being able to figure out the distribution of the total intensity of pollutants based on their activity becomes evident, as this method can make them aware of the real dimension of the problem and avoid classical myths, like the idea that most of the pollution is acquired outdoors (see Figure 6). While there are similar studies such as in [87], where the authors demonstrated the cleanest air routing algorithm for path navigation by calculating the PM${}_{2.5}$ exposure, they mainly focused on pollution acquired outdoors and not indoors.

Since information is the key aspect in having the opportunity to make proper decisions, the advantage of such an integrated framework that is able to integrate not only outdoor conditions but also indoor ones when available becomes more evident. This can also have an impact not only at the individual level by making everyone aware of their exposed pollution levels but at an aggregated level as well, because the public health dimension is impacted when buildings are seen as actionable regarding the indoor conditions. Therefore, KPIs can be adopted by considering the gradient between outdoor and indoor levels per area of occupancy of the buildings. By having systematic monitoring inside, the management dimension can be adopted.

### 4.4. Alternative Travel Routes

Similar to [87,94], another non-neglectable dimension that is possible to consider is the impact in terms of transportation decisions. Figure 9 shows different routes that one individual can take when moving from one location to another, and the corresponding aggregated pollution and exposure values are provided in Table 6. These values were predicted on the basis of the individual’s activity data for 24 March 2017 from 12:04:30 to 12:23:44. The most frequent one adopted by the user was labeled “Actual”, while the other potential routes were named A to C.

In this example, the better individual route will is B, as it causes the least amount of PAPE at 0.769 μg, which is 22.75% lower than the actual exposure of 0.995 μg. However, the decision process can be more complex, because there will certainly be some time duration uncertainties, which will consequently result in uncertainty about the total PAPE value of each alternative route.

Although most of the tools that give routing solutions for transportation problems are based on duration, some of them have the capability of filtering them out based on pollution exposure outdoors [87,94]. In terms of added value, this contribution enables alternatives to be ranked based on estimated pollution levels both outdoors and indoors, provided that pollution data is also available inside public transportation modes such as trains, buses, and subways. In these cases, as forecast for pollution is needed, machine-learning-based models that infer outdoor pollution values need to be used.

### 4.5. Limitations

Due to the lack of publicly available air quality information for other indoor areas such as shops, buses, cars, metros, etc., outdoor pollution data from the fixed-site outdoors information must be used in such cases. If there are more available resources, additional monitoring IoT devices in other indoor areas will provide greater accuracy. In most cases, good results demand good inputs, and existing data are replaced whenever better data become available. Quality improvements can be expected from those actions. Smart city empowered data sharing platforms such as IOTA Tangle [95] would boost IoT-based indoor air quality resource availability.

Accuracy for outdoor pollution estimation is another known limitation, both because of the time frequency resolution of available data and because of the interpolation errors. It would be possible to implement Weather Research and Forecasting (WRF) models such as the CMAQ. This decision requires significant effort, not only because of using the appropriate Digital Elevation Model (DEM) required to represent the landscape and building configuration, which is a complex task, but because it requires the boundary conditions to be realistic. This means adopting pressure and wind speed conditions for all surfaces external to the volume of interest. These situations need to be updated regularly throughout the day, as environmental conditions change as well. Indeed, numerical stability conditions must be carefully managed in this case as well.

For future applications, the best solution for environments will come from both the increasing deployment of dense (e.g., street-level) IoT-based air quality sensors and the prosperity of the data sharing platform, which can increase the available data and, consequently, will increase the accuracy.

## 5. Conclusions

This paper (1) proposed a framework for an AQDSS and (2) developed an IoT application based on this framework. The feasibility of the IoT application in measuring PAPE was evaluated through a case study. In comparison to mobile sensors that were used in previous studies, this IoT application has higher scalability, because it involves minimal cost and intrusion to the individual. This pilot case study also presents evidence that PAPE can be estimated by employing indoor monitors and e-beacon technologies that have not been used previously in similar studies.

Using our proposed framework as a general guideline, the IoT application that we developed can be further extended to include prediction models that will allow an individual to make smart decisions when it comes to PAPE risk. Furthermore, PAPE data obtained from the application can be used in air quality policy development as well as in epidemiological studies to explore the correlations of PAPE with certain diseases.

We faced difficulties during the extraction and integration of data from multiple devices, which highlights the importance of choosing the right technologies to use when developing such IoT applications. There was an observed variance among the different devices, which can be attributed to device calibration and the measurement techniques used. Future research should, therefore, explore these issues and identify emerging technologies that permit seamless data integration and more accurate PAPE measurements.

## Figures and Tables

**Figure 1 ijerph-16-03130-f001:**
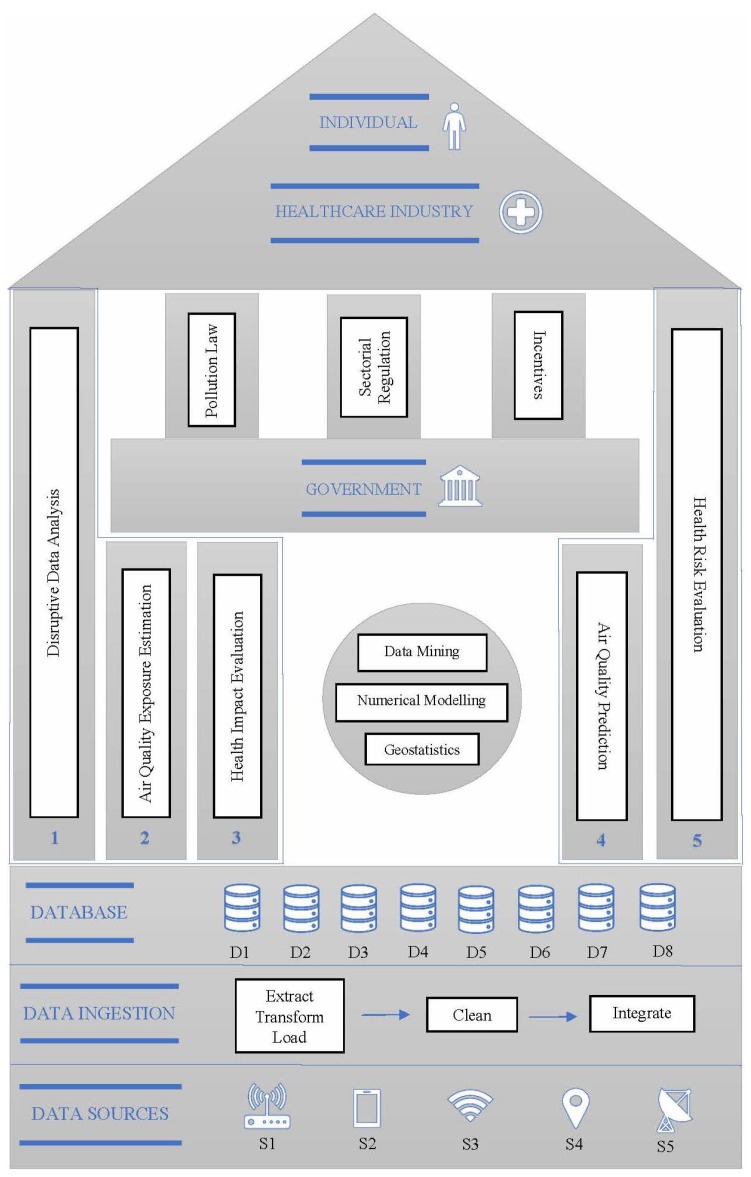
Proposed framework for the Air Quality Decision Support System (AQDSS).

**Figure 2 ijerph-16-03130-f002:**
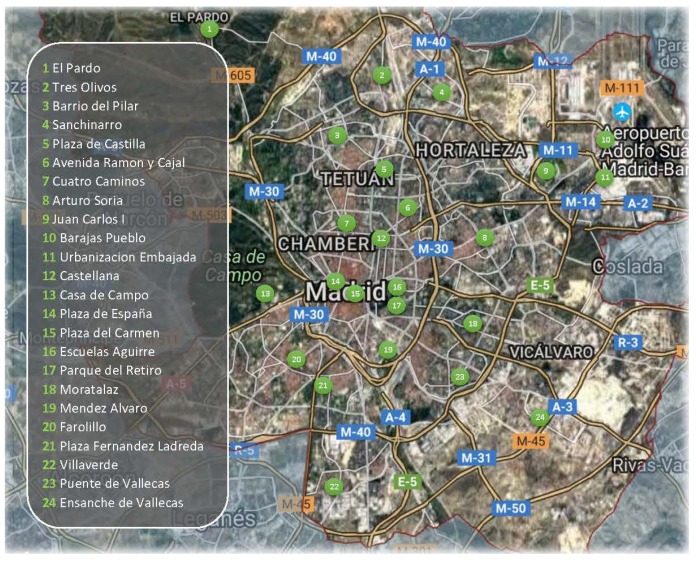
Air Quality Monitoring Network of Madrid.

**Figure 3 ijerph-16-03130-f003:**
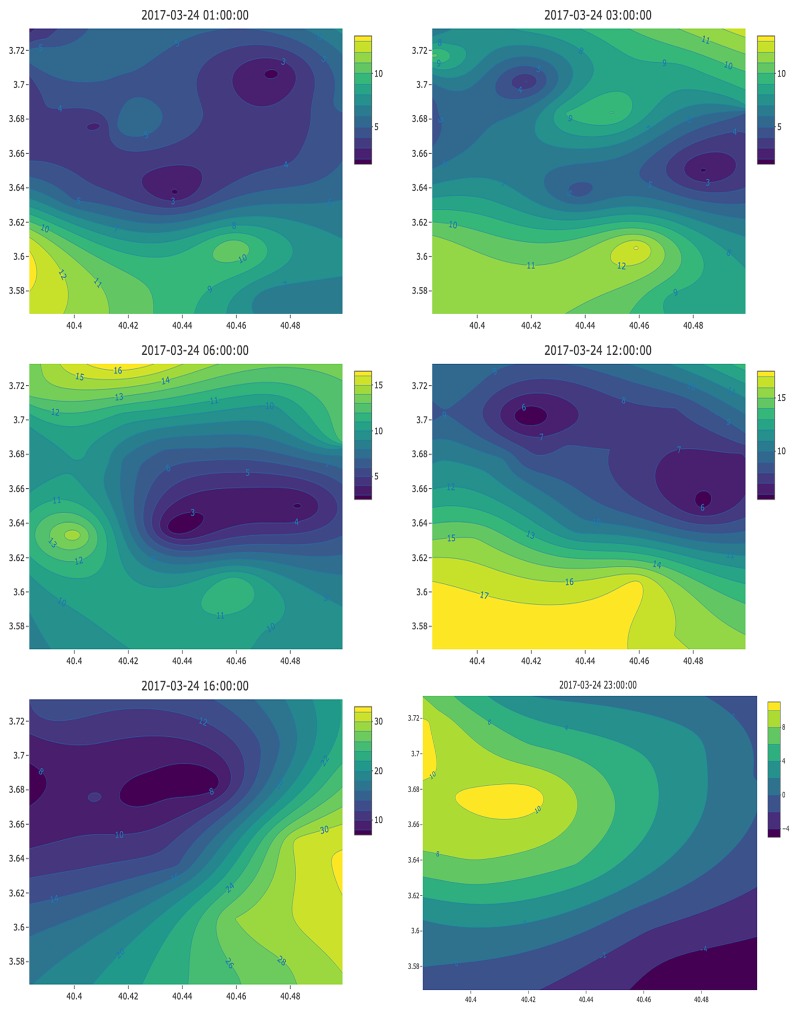
Co-kriging interpolation of PM${}_{2.5}$ on 24 March 2017.

**Figure 4 ijerph-16-03130-f004:**
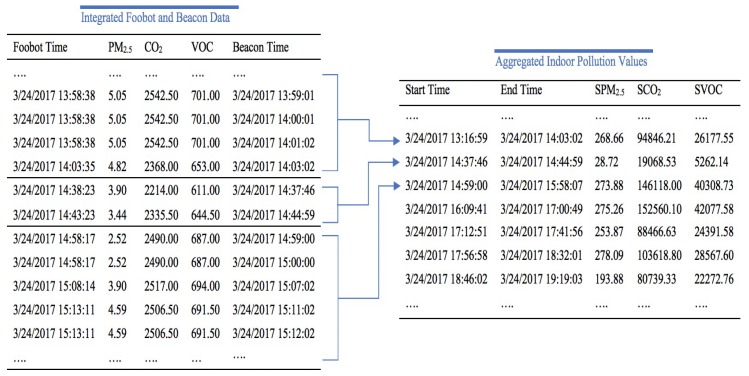
Aggregation of Indoor Pollution Values.

**Figure 5 ijerph-16-03130-f005:**
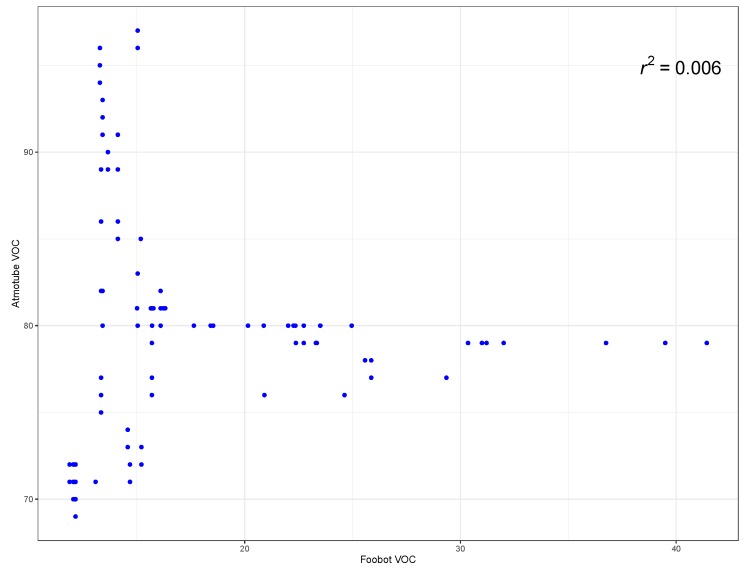
Indoor VOC Values from Atmotube and Foobot for 2017-03-31.

**Figure 6 ijerph-16-03130-f006:**
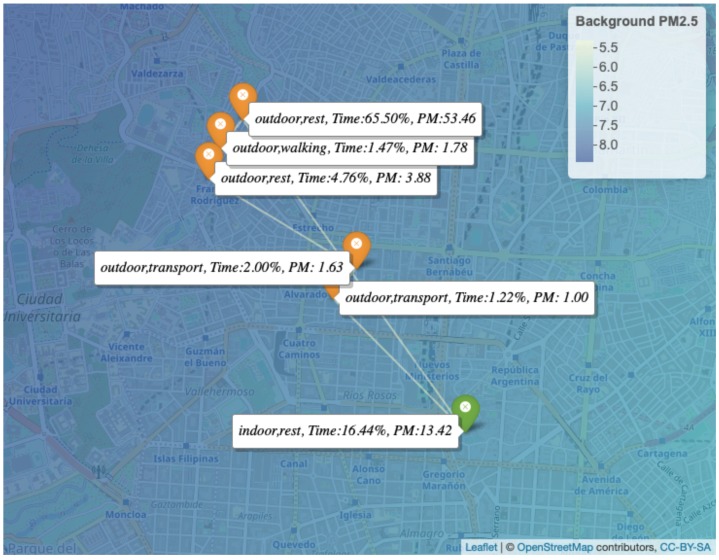
One Day PM${}_{2.5}$ Exposure Per Location, Activity Type, and Time Percentage.

**Figure 7 ijerph-16-03130-f007:**
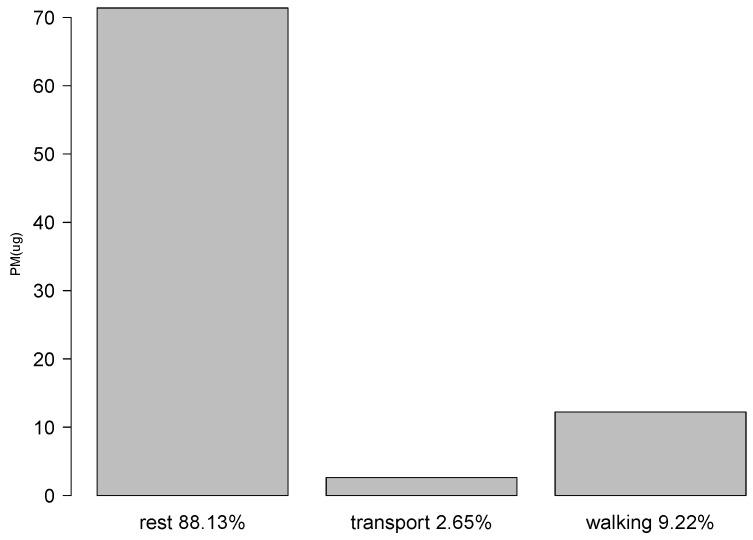
One Day PM${}_{2.5}$ Exposure by Activity Type Percentage.

**Figure 8 ijerph-16-03130-f008:**
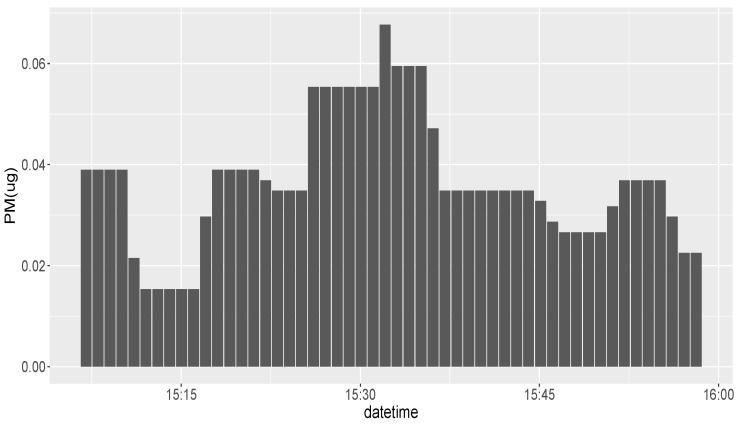
Indoor PM${}_{2.5}$ Exposure Values Across Time. (DeltaT = 1 min).

**Figure 9 ijerph-16-03130-f009:**
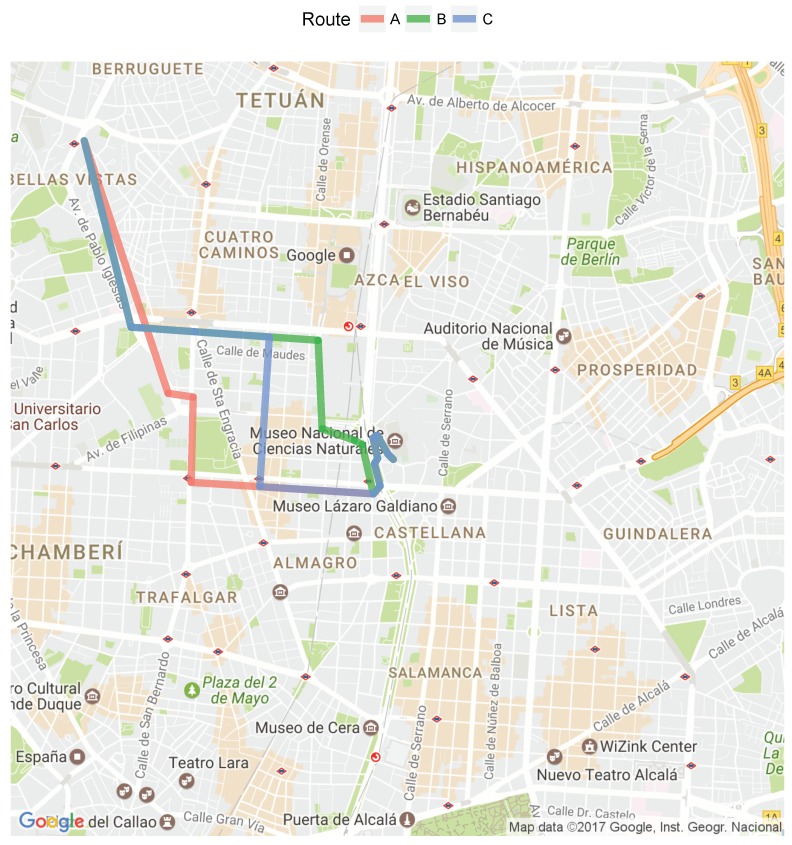
Alternative Travel Routes.

**Table 1 ijerph-16-03130-t001:** Available Pollutants for each Data Source.

Source	Pollutant	Unit
	PM${}_{2.5}$	μg/m${}^{3}$
Indoor Monitor	CO${}_{2}$	ppm
	VOC	ppb
	PM${}_{2.5}$	μg/m${}^{3}$
	PM${}_{10}$	μg/m${}^{3}$
	CO	μg/m${}^{3}$
Outdoor Monitor	NO${}_{2}$	μg/m${}^{3}$
	SO${}_{2}$	μg/m${}^{3}$
	O${}_{3}$	μg/m${}^{3}$
	NO${}_{x}$	μg/m${}^{3}$

**Table 2 ijerph-16-03130-t002:** Techniques Used for Outdoor Pollution Modeling.

Technique	IDW	Simple Kriging	Ordinary Kriging	Co-Kriging
**Formula**	$\hat{z_{0}}=\sum_{i=0}^{n}\lambda_{i}X_{i}$	$\hat{z_{0}}-c=\sum_{i=0}^{n}\lambda_{i}\left(X_{i}-c\right)$	$\hat{z_{0}}=\sum_{i=0}^{n}\lambda_{i}X_{i}$	$\hat{z_{0}}=\sum_{i=0}^{n}\lambda_{i}X_{i}+\sum_{j=0}^{n}\beta_{j}t_{j}$
**Characteristics**	The weight, $\lambda_{i}$, depends solely on the distance to the prediction location.	Assumes a constant and known mean c of the samples. The weight, $\lambda_{i}$, depends on the use of a fitted model to the measured points, the distance to the prediction location, and the spatial relationships among the measured values around the prediction location.	Condition that $\sum_{i=0}^{n}\lambda_{i}=1$ assumes a constant and unknown mean of the samples. The weight, $\lambda_{i}$, depends on the use of a fitted model to the measured points, the distance to the prediction location, and the spatial relationships among the measured values around the prediction location.	$t_{j}$ is the secondary regionalized variable which is co-located with the target variable $t_{j}$. The weight $\beta_{j}$ assigned to $t_{j}$ varies between 0 to 1.
**Parameters**	Idp 0.1,0.3,0.5,0.8,2,5	Sill 1,10,100,250,500,600,700,800,900
		Range 0.1,0.5,1,10,20,50,80
		Nugget 0.00001,0.0001,0.001,0.01,0.1,1,10,100,200,50
		Beta 0.05,0.12,0.2,0.5,0.9,1.5,3,1
		Variogram Model: Gaussian, Circular, Exponential

**Table 3 ijerph-16-03130-t003:** Available Pollutants for each Data Source.

Model	Function
Circular	$semivar\left(d\right)=c_{0}+c(1-\frac{2}{\pi }cos^{-1}\left(\frac{d}{\alpha }\right)+\sqrt[{}]{1-\frac{d^{2}}{\alpha^{2}}}$ 0<d≤α
	$semivar\left(d\right)=c_{0}+cd>\alpha$
	sermivar(0)=0
Spherical	$semivar\left(d\right)=c_{0}+c(\frac{3d}{2\alpha }-\frac{1}{2}{\left(\frac{d}{\alpha }\right)}^{3}$ 0<d≤α
	$semivar\left(d\right)=c_{0}+cd>\alpha$
	sermivar(0)=0
Exponential	$semivar\left(d\right)=c_{0}+c\left(1-e^{\frac{-d}{r}}\right)$ d>0
	sermivar(0)=0
Gaussian	$semivar\left(d\right)=c_{0}+c\left(1-\exp\left(\frac{-d^{2}}{r^{2}}\right)\right)$ d>0
	sermivar(0)=0

*d* = distance between two locations, $c_{0}$ = y-intercept, α = range.

**Table 4 ijerph-16-03130-t004:** Data from Tracking App.

Start Time	End Time	Latitude	Longitude	Activity
2017-03-24 00:00:00	2017-03-24 11:55:37	40.4612	−3.7093	Rest
2017-03-24 11:55:37	3/24/2017 11:59:20	40.4592	−3.7106	Walk
2017-03-24 11:59:20	3/24/2017 12:04:30	40.4571	−3.7118	Rest
2017-03-24 12:04:30	3/24/2017 12:23:44	40.4486	−3.7006	Transport
2017-03-24 12:23:44	3/24/2017 19:52:00	40.4400	−3.6894	Rest
2017-03-24 19:52:00	3/24/2017 20:08:40	40.4506	−3.6994	Transport
2017-03-24 20:08:40	3/24/2017 21:13:07	40.4612	−3.7093	Rest
2017-03-24 21:13:07	3/24/2017 21:21:55	40.4594	−3.7105	Walk
2017-03-24 21:21:55	3/24/2017 22:32:59	40.4575	−3.7117	Rest
2017-03-24 22:32:59	3/24/2017 22:42:51	40.4594	−3.7105	Walk
2017-03-24 22:42:51	3/25/2017 00:00:00	40.4612	−3.7093	Rest

**Table 5 ijerph-16-03130-t005:** Integrated Indoor and Outdoor Personal Air Pollution Exposure (PAPE).

Start	End	Latitude	Longitude	${\mathrm{PM}}_{2.5}$	Environment	Activity	VE	Exposure
				(μg/m${}^{3}$ * min)			(m${}^{3}$/min)	(μg)
2017-03-24 0:00	2017-03-24 11:55	40.461	−3.709	5354.15601	Outdoor	Rest	0.00893	47.81261
2017-03-24 11:55	2017-03-24 11:59	40.459	−3.711	13.9543	Outdoor	Walk	0.01326	0.18503
2017-03-24 11:59	2017-03-24 12:04	40.457	−3.712	20.08333	Outdoor	Rest	0.00893	0.17934
2017-03-24 12:04	2017-03-24 12:23	40.449	−3.701	111.43482	Outdoor	Transport	0.00893	0.99511
2017-03-24 13:16	2017-03-24 14:03	40.43999	−3.68938	268.65655	Indoor	Rest	0.00893	2.3991
2017-03-24 14:03	2017-03-24 14:37	40.44	−3.689	347.06125	Outdoor	Walk	0.01326	4.60203
2017-03-24 14:37	2017-03-24 14:44	40.43999	−3.68938	28.72532	Indoor	Rest	0.00893	0.25652
2017-03-24 14:44	2017-03-24 14:59	40.44	−3.689	141.90761	Outdoor	Walk	0.01326	1.88169
2017-03-24 14:59	2017-03-24 15:58	40.43999	−3.68938	273.87957	Indoor	Rest	0.00893	2.44574
2017-03-24 15:58	2017-03-24 16:09	40.44	−3.689	80.2915	Outdoor	Walk	0.01326	1.06467
2017-03-24 16:09	2017-03-24 17:00	40.43999	−3.68938	275.25632	Indoor	Rest	0.00893	2.45804
2017-03-24 17:00	2017-03-24 17:12	40.44	−3.689	70.15613	Outdoor	Walk	0.01326	0.93027
2017-03-24 17:12	2017-03-24 17:41	40.43999	−3.68938	253.86892	Indoor	Rest	0.00893	2.26705
2017-03-24 17:41	2017-03-24 17:56	40.44	−3.689	63.02457	Outdoor	Walk	0.01326	0.83571
2017-03-24 17:56	2017-03-24 18:32	40.43999	−3.68938	278.09301	Indoor	Rest	0.00893	2.48337
2017-03-24 18:32	2017-03-24 18:46	40.44	−3.689	85.27723	Outdoor	Walk	0.01326	1.13078
2017-03-24 18:46	2017-03-24 19:19	40.43999	−3.68938	193.88363	Indoor	Rest	0.00893	1.73138
2017-03-24 19:52	2017-03-24 20:08	40.451	−3.699	182.9052	Outdoor	Transport	0.00893	1.63334
2017-03-24 20:08	2017-03-24 21:13	40.461	−3.709	626.14768	Outdoor	Rest	0.00893	5.5915
2017-03-24 21:13	2017-03-24 21:21	40.459	−3.711	68.61289	Outdoor	Walk	0.01326	0.90981
2017-03-24 21:21	2017-03-24 22:32	40.457	−3.712	414.90097	Outdoor	Rest	0.00893	3.70507
2017-03-24 22:32	2017-03-24 22:42	40.459	−3.711	51.34302	Outdoor	Walk	0.01326	0.68081
2017-03-24 22:42	2017-03-25 0:00	40.461	−3.709	6.56381	Outdoor	Rest	0.00893	0.05861

**Table 6 ijerph-16-03130-t006:** Total PM${}_{2.5}$ and Exposure Values of the Different Routes..

Route	${\mathrm{PM}}_{2.5}$ (μg/m${}^{3}$ * min)	Exposure (μg)
A	89.34	0.80
B	86.08	0.77
C	100.17	0.89
Actual	111.43	0.99

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
