# Peer review of "The Use of the Internet of Things for Estimating Personal Pollution Exposure"

_ijerph, 2019, doi:10.3390/ijerph16173130_

Round 1

Reviewer 1 Report

The topic of this paper is interesting, and I was triggered by the title of the manuscript. However, the paper did not fulfill my expectations. Because of the unconventional structure of the paper, it is difficult to follow. The methods used in the framework (sensors, air pollution models) are not state-of-the-art, and the case study only considers 1 person. This is a pity because there are some nice elements in the paper such as the use of the beacon technology for indoor pollution, the use of the Moves app, or the general framework as proposed in Figure 1. I would prefer this paper to be split up in two papers: one discussing the framework in a more theoretical fashion, and one discussing the case-study however you would need to improve the methods for this and need more measurement data.

General comments:

-          The abstract overstates the reach of the study. The case study consists of 1 person only which is a pity given the statement that the big advantage of your approach is scalability.

-          The models used are not state-of-the-art (kriging, IDW). The same holds for the air quality sensors. The validation of the sensors seems not very good (Figure 4, but no R² provided), and as a standard practice the quality of low-cost sensors should be thoroughly checked before being used in a field study. The overall bad performance of all air pollution sensors in field studies is completely ignored.

-          The literature review is not necessary in this paper. It should be either shortened and merged with the introduction, or moved to an appendix, or removed. The review is non-systematic and the added value to the paper is low.

-          Please reduce the number of references; several references refer to similar research and can be cut. Contradictory to this, the results from the case study are not or barely compared to the literature.

-          The overall layout should be improved. Some sentences are in red or green and it is unclear what this means. The numbering of the figures is inconsistent, and the references to tables or figures in the text is sometimes showing an error message. The text on Figure 5 (map) is illegible. For this draft version, please insert tables and figures where they should be placed in the text, and not before of several pages later (and with formulas in between) – it makes it very hard to read.

Specific comments:

-          Paragraph 2.2: The title of this section is not describing what is presented in the text.

-          Line 147-148: This is a generalization, and not valid for all methods, for example for land use regression models.

-          Line 173-174: Most of the indoor air penetrates from outdoors.

-          Line 190-193: GPS that does not function indoors can be seen as an opportunity – this way it is perfectly possible to identify when an individual is indoors. The exact room the person is in, is less important for air pollution exposure assessment.

-          Line 246-249: This is not understandable in this context.

-          Section 3.2.1: This section on the study area is too detailed.

-          Line 321: Isn’t the home location the indoor location where someone spends most of his time?

-          Lines 332-334: I applaud the efforts of the authors to share their code on Zenodo.

-          Lines 345-348: I understand that simplicity and computational costs are factors in the decision to choose one model over another, however the model needs to be sufficiently good and accurate and this should be the number one priority. When you choose a less accurate model, the added value of personal exposure assessment is barely zero because then the uncertainty of your estimate will be larger than differences between individuals.

-          Formula 5: It doesn’t really matter, but why ‘SZ’? Is it an abbreviation?

-          Section 3.2.8: Though interesting, it is unclear to me how this section fits within this paper.

-          The Moves app can give you more exact tracking data (along the route). Why is this data not used?

-          Figure 4 (Indoor VOC values from Atmotube and Foobot): This graph is not useful. Please use a correlation plot.

-          Figure 5 (One day PM2.5 exposure per location, activity type, and time percentage): This figure needs to be improved: (1) the text is nearly illegible; (2) the caption needs to be much clearer; (3) the overall resolution of the figure needs to be higher; (4) add units for PM concentrations; (5) add a scale. The home location is classified as ‘outdoor, rest’ – isn’t that a bit weird? This is the most important results figure of the case-study; it needs to be more attractive.

-          Table 6 and the discussion of this table is not useful and should be removed.

-          Figure 8: Can you also show the actual route on the map?

-          Line 533: The ranking of alternative routes is presented as something new, but in fact applications doing this already exist. Refer to literature instead of presenting it as something innovative.

-   Line 553: I don’t agree with this statement. Distributing large amounts of bad sensors over a city will not result in an accurate estimate – so large numbers do not automatically imply good accuracy.

Author Response

The topic of this paper is interesting, and I was triggered by the title of the manuscript. However, the paper did not fulfill my expectations. Because of the unconventional structure of the paper, it is difficult to follow. The methods used in the framework (sensors, air pollution models) are not state-of-the-art, and the case study only considers 1 person. This is a pity because there are some nice elements in the paper such as the use of the beacon technology for indoor pollution, the use of the Moves app, or the general framework as proposed in Figure 1. I would prefer this paper to be split up in two papers: one discussing the framework in a more theoretical fashion, and one discussing the case-study however you would need to improve the methods for this and need more measurement data

Many thanks for your valuable comments. We restructured the paper in our updated manuscript and make it more readable.  Firstly, we have to emphasize that we fully consider, understand and agree with your concerns. As we highlight in the paper, it is a conceptual (theoretical) paper according to our view, which allow us to produce a referential framework that we show it becomes valid throughout a single case. And this can be seen just as a pilot example. For sure, we are considering to produce a more practical and extensive paper covering several application cases, fitting inside the previous framework. However, as it involves different cases with different configurations, people, contexts, etc., it will last for several months. To be consistent with the dissemination strategy we will consider IJERPH for the next one as well.

General comments:

-          The abstract overstates the reach of the study. The case study consists of 1 person only which is a pity given the statement that the big advantage of your approach is scalability.

Thank you very much for your opinion. We would like to point out that the current version is a conceptual paper in which a theoretical framework is promoted and a case study was made as a proof of concept to demonstrate the practical implications of the proposed IoT application. We have revised the abstract accordingly to include this statement. It is not our intention to overstate the reach of the study, we are merely trying to show the possible future practical applications where our proposed framework and IoT application can be used.

We are aware of the limitations of including only 1 person in our case study. However, since this study is mainly interested in estimating the pollution exposure of an individual, concentrating on a single volunteer provides a fine-grained detail on the associated issues related to the technical feasibility of our proposed application. For example, in the paper by Su, J.G. et al., 2015. Integrating smart-phone based momentary location tracking with fixed site air quality monitoring for personal exposure assessment. Science of The Total Environment, 506–507, pp.518–526. Available at: //www.sciencedirect.com/science/article/pii/S0048969714016040, the authors argue that “since we are interested in extracting routine behaviors of individual users we focused on long-term, fine grain data acquisition from a few users rather than on coarse data from a large user population…Focusing on a single person over three months of time helped us identify the issues associated with smart phone momentary location tracking in very fine detail.”  Of course, in future work, we will try to increase the number of users so we can derive more robust conclusions about our proposed method.

-          The models used are not state-of-the-art (kriging, IDW).

Thank you very much indeed for your opinion. Actually, we did our best in providing a consistent perspective about how to model outdoor pollution, including the four major classes (see section 3.1.2). Indeed, we provided a rather agnostic justification about the choice, including the potential added value due to data fusion, which can enable machine learning based models.

In addition, in section 4.1 potential limitations like roughness are discussed and some other limitations from other techniques like boundary conditions for CMAQ are also discussed.

We do believe we have argued to explain why the specific class of models has been chosen as well as their limitations. To this end, it makes sense to clarify that the contribution of the paper is not to promote the more accuracy model but to provide an integrated approach, able to be ran dynamically and providing timely information to the users. Based on these aspects, we cannot agree on the sentence about the state-of-the-art models raised by the reviewer.

- The same holds for the air quality sensors. The validation of the sensors seems not very good (Figure 4, but no R² provided), and as a standard practice the quality of low-cost sensors should be thoroughly checked before being used in a field study. The overall bad performance of all air pollution sensors in field studies is completely ignored.  

As you suggested, we updated the Figure 4. Regarding to the low-cost sensors in the field studies, Regarding the low-cost sensors in field studies, there are several papers that already provide a useful information about cost, time resolution etc., both from individual low-cost sensors and network of sensors like  Yi, W., Lo, K., Mak, T., Leung, K., Leung, Y., & Meng, M. (2015). A survey of wireless sensor network based air pollution monitoring systems. Sensors, 15(12), 31392-31427. There are also papers discussing quality of measurement process and power used like Yi, W. Y., Leung, K. S., & Leung, Y. (2018). A modular plug-and-play sensor system for urban air pollution monitoring: design, implementation and evaluation. Sensors, 18(1), 7. We didn't spend much more effort in this direction as our paper aims to be closer to the value creation from the information side and because we already have a relevant number of references. However, we could include such discussion if the review considers it relevant.

-          The literature review is not necessary in this paper. It should be either shortened and merged with the introduction, or moved to an appendix, or removed. The review is non-systematic and the added value to the paper is low.

We did a thorough literature review based on the previous comments from the reviewers and in our humble opinion, we do believe that it is a necessary part of our paper since it supports the choices that we made in terms of the models that we used for our analysis (section 2.3) as well as the air pollutants that we have considered (section 2.1). It also discusses air pollutant determination method (section 2.2) and the gaps in current literature which we are trying to address with our paper (section 2.4).

-          Please reduce the number of references; several references refer to similar research and can be cut. Contradictory to this, the results from the case study are not or barely compared to the literature.

Although previous reviewers have asked us to add a few references to make the literature review more comprehensive, we agree that references referring to similar research can be cut and hence, we have now cut down such references keeping only the most recent and relevant ones. We have also added some references in the case study section of the paper which compares our work to existing studies (lines 349-350, 387-388, 445, 467-459, 463-465, 488-490, 516-518, 528-529, 538-542).

-          The overall layout should be improved. Some sentences are in red or green and it is unclear what this means. The numbering of the figures is inconsistent, and the references to tables or figures in the text is sometimes showing an error message. The text on Figure 5 (map) is illegible. For this draft version, please insert tables and figures where they should be placed in the text, and not before of several pages later (and with formulas in between) – it makes it very hard to read.

We apologize for the confusion on the layout. The colored sentences were the changes that we have made based on the comments from the previous reviewers. We have now updated the layout including the tables, figures, and references so that they are consistent and readable.

Specific comments:

-          Paragraph 2.2: The title of this section is not describing what is presented in the text.

Thank you for your constructive comments for improving our work. We agree the title could not well reflect the whole section picture; we have changed to [Determination Method for Air Pollutants].

-          Line 147-148: This is a generalization, and not valid for all methods, for example for land use regression models.

Thank you for the opinion of the reviewer. We can refer to different papers like Xiangyu Jiang, Eun-hye Yoo, “The importance of spatial resolutions of Community Multiscale Air Quality (CMAQ) models on health impact assessment”, Science of The Total Environment, Volume 627, 2018, Pages 1528-1543, ISSN 0048-9697, https://doi.org/10.1016/j.scitotenv.2018.01.228. (http://www.sciencedirect.com/science/article/pii/S0048969718302699). There, you can see how when CMAQ is simulated in a supercomputer center the computation cost for a mesh with a side of 36Km of side is 300 times less than the cost when the mesh has 2Km side. Indeed, the recommendation is to use a mesh of 12 Km in order to have acceptable figures in such 32 core with 256Gb of RAM computing farm.

From the paper Ádám Leelőssy, István Lagzi, Attila Kovács, Róbert Mészáros, “A review of numerical models to predict the atmospheric dispersion of radionuclides”, Journal of Environmental Radioactivity, Volume 182, 2018, Pages 20-33, ISSN 0265-931X, https://doi.org/10.1016/j.jenvrad.2017.11.009,  you can find  “On the other hand, microscale simulations (on the scale of 10–1000 m) are based on the detailed solution of the flow and concentration field among buildings and other surface obstacles. These simulations have extreme hardware requirements comparable to or even larger than global scale problems.“

Based on such evidences and many more the authors believe that by saying “However, the expensive computational cost and failure to capture pollution variability make them inadequate for the application of modelling in REAL TIME in urban areas (microscale) where there are severe photochemical pollution conditions.“ is not a generalization. We do not claim that every single model can be applicable, at all. You can find a review about applicability conditions in the latest reference provided, which mainly agrees with our message.

-          Line 173-174: Most of the indoor air penetrates from outdoors.

Thank you for your view. Actually, indoor exposure levels account for several aspects. Firstly, diffusion of outdoor air, it depends on elements such as ventilation and indoor and outdoor temperature gradient. However, there are the other primary indoor pollutant sources such as building materials, furnishings, cleaning products, tobacco products, besides, indoor pollutants can also be produced from human activities (e.g. occupancy) indoors, industrial manufacturing process of corresponding work patterns. As introduced in text line 38-40, indoor air pollutants may occasionally be more than 100 times higher than outdoors.

-          Line 190-193: GPS that does not function indoors can be seen as an opportunity – this way it is perfectly possible to identify when an individual is indoors. The exact room the person is in, is less important for air pollution exposure assessment.

Thank you for your perspective. Variability across microenvironments in the same indoor environment have been investigated by numerical research work. They demonstrated significant temporal and spatial variations in different microenvironments.  Corresponding to relevant references: 

smoking and no-smoking areas.  Jayes LR, Ratschen E, Murray RL, Dymond-White S, Britton J. Second-hand smoke in four English prisons: an air quality monitoring study. BMC Public Health. 2016;16:119. Published 2016 Feb 4. doi:10.1186/s12889-016-2757-y.

underground and ground-level in metro system.  Cartenì, Armando & Cascetta, Furio & Campana, Stefano. (2015). Underground and ground-level particulate matter concentrations in an Italian metro system. Atmospheric Environment. 101. 328-337. 10.1016/j.atmosenv.2014.11.030.

office and photocopy room. Fathallah, Houssem Eddine et al. “An IoT-based scheme for real time indoor personal exposure assessment.” 2016 13th IEEE Annual Consumer Communications & Networking Conference (CCNC) (2016): 323-324.

Besides, in household, although people spent less time in the kitchen than other rooms, but cooking preparation process takes place mostly using electric or gas cookers, or leaky stove in developing countries such as China and India.  Those could lead people exposure into high concentration of indoor pollutants than other rooms.

Therefore, to differentiate microenvironments indoors, as shown in those applications mentioned above, has a significant implication for indoor pollution exposure assessment in terms of quality assurance and assessment accuracy.

-          Line 246-249: This is not understandable in this context.

Actually, to provide these details was requested by a previous review of the paper. The authors agree that the provided technical details can be considered as not necessary. We can remove them, but they were placed upon demand. Anyhow, a better wording is presented in this version, trying to accommodate the explanation in a more suitable way.

-          Section 3.2.1: This section on the study area is too detailed.

The details information was indeed required by the previous reviewer. However, based on your recommendation, we tried to keep the key information and reduce some parts in the section line (304-309 in old version).

-          Line 321: Isn’t the home location the indoor location where someone spends most of his time?

Actually, our volunteer spends most of his time in the workplace in our experiment time period.

-          Lines 332-334: I applaud the efforts of the authors to share their code on Zenodo

          Thank you!

-          Lines 345-348: I understand that simplicity and computational costs are factors in the decision to choose one model over another, however the model needs to be sufficiently good and accurate and this should be the number one priority. When you choose a less accurate model, the added value of personal exposure assessment is barely zero because then the uncertainty of your estimate will be larger than differences between individuals

Thank you for the provided perspective. In the humble opinion of the authors, the point raised by the reviewer is very relevant. We must be aware about different uncertainty sources for both indoor and outdoor measurements. For sure, one of them is due to the outdoor modeling of pollution but it is not the largest one, and there are estimations for their accuracy. The paper does not claim that it proposes a full accurate estimation, but an estimation trying to promote feasible and integrated solution to estimate the personal pollution dosis. Such dosage is composed by different components, both from outdoor and indoor.

Therefore, the uncertainty based on the outdoor model will impact in the same way to two different individuals being outdoor at the same point in time. Indeed, data fusion will contribute to increase the precision specifically for interpolated based models.

Finally, yet importantly, moderate uncertainty from the concentration of the pollution levels need to be integrated through the exposition time according to the air ingestion rate (another source of uncertainty, depending of the physical activity being carried out). Because of those moderating factors the total figure of uncertainty remains much less relevant that the aggregated exposition figure.

-          Formula 5: It doesn’t really matter, but why ‘SZ’? Is it an abbreviation?

            SZ accounts for the level of pollution in a particular space for a period of time. As Z was the abbreviation for pollution level, we have named SZ for "Summed Up Z" the aggregated values in the relevant period of time named p.

-          Section 3.2.8: Though interesting, it is unclear to me how this section fits within this paper.

This section illustrates the possible practical application or use-case in which our proposed framework and IoT application can be used. We have changed the title of the section to “Practical Application”, to better reflect the purpose of this section.

-          The Moves app can give you more exact tracking data (along the route). Why is this data not used?

Indeed, the route A is the actual path for the user by using the data from Moves app. Nevertheless, the temporal resolution for Moves app is not high, making it hard to have more exact tracking data.

-          Figure 4 (Indoor VOC values from Atmotube and Foobot): This graph is not useful. Please use a correlation plot. 

Thank you, as you recommended, we have updated the plot.

-          Figure 5 (One day PM2.5 exposure per location, activity type, and time percentage): This figure needs to be improved: (1) the text is nearly illegible; (2) the caption needs to be much clearer; (3) the overall resolution of the figure needs to be higher; (4) add units for PM concentrations; (5) add a scale. The home location is classified as ‘outdoor, rest’ – isn’t that a bit weird? This is the most important results figure of the case-study; it needs to be more attractive.

-          Table 6 and the discussion of this table is not useful and should be removed.

Thank you for the recommendation, we removed this table in our updated manuscript. 

-          Figure 8: Can you also show the actual route on the map?

As we discussed in your previous comments, route A indeed is the actual path.

-          Line 533: The ranking of alternative routes is presented as something new, but in fact applications doing this already exist. Refer to literature instead of presenting it as something innovative. 

We have revised this part by referring to similar studies in literature and pointing out that as opposed to such applications, our approach also considers indoor pollution (waiting time inside subways for instance) when calculating the travel routes.  Since indoor pollution data is currently not available in public transportation such as buses, trains, and subways (which we have also mentioned in section 4.5), we were not able to fully test such approach in our case study. Nevertheless, in our proposed framework, we acknowledged that indoor pollution is an integral part of estimating PAPE for such alternative travel routes.

-   Line 553: I don’t agree with this statement. Distributing large amounts of bad sensors over a city will not result in an accurate estimate – so large numbers do not automatically imply good accuracy.

Thank you for your comment. Although the authors understand the reviewer’s concern, they can’t endorse the IoT based sensors are ‘bad sensors’. They are not class 0 sensors (or at least they do not need to be) but having a standard quality the benefit of a dense mesh of sensors becomes evident. Such strategy, as described in the paper (see section 3.1.2) is reported as being done in different cities around the world.  

Reviewer 2 Report

I appreciate the opportunity to review this paper which aims at estimating the personal pollution exposure. The case study employed is from Madrid, Spain. It seems this paper is the revised version of an original submission. The revised version has been improved substantially and merits publication at this stage. I do not have any comments as I feel the authors have addressed and incorporated them very satisfactorily. 

Author Response

Thank you so much for your time and providing valuable comments to improve this manuscript.

This manuscript is a resubmission of an earlier submission. The following is a list of the peer review reports and author responses from that submission.

Round 1

Reviewer 1 Report

This paper focuses on “Internet of Things for Estimating Personal Pollution Exposure” and the topic is interesting. The paper needs editing in terms of English and scientific writing style and some parts of the paper are misleading or have no logical order or flow. The paper must go through editing and proofreading process before publication and strictly follow the Guideline for the Author(s). The manuscript has been very casually and carelessly prepared.

The ABSTRACT needs to be revised and special attention should be taken in Language. Moreover, it should be written in proper flow by following; basic introduction, problem, the methodology adopted and results.

One of the limitations of the paper is that it lacks critical related work but it can be improved before publication. The proposed study is not critically evaluated and compared to the related work/state-of-the-art and is not identified and discussed its drawbacks and limitations. Thus, it is not easy to assess the real contribution of the paper in the field and how much is efficient the proposed solution compared to related works. A clear assessment of the contribution of the authors when compared to existing approaches should be given.

At the end of the literature review, you should have a summary of what the cited literature implies. This summary can be a paragraph to bring together all the pieces of literature you reviewed and to clarify where your research stands. In addition, I feel that additional and more recent works are needed to be examined.

The reported results are interesting but they can be polished by adding more explanation.

The Development Issues are not discussed so it needs to be better described.

A better discussion and analysis should be made. What exactly it denotes? Why it is important?

The reference and citation formats are wrong and most importantly most of the references and figures are not cited with proper captions.

Overall, the manuscript in the current form is not up to the standard as per the novelty and standards of the MDPI journal. So, in my view, the manuscript should be seriously rewritten in both research design and writing style.

Author Response

This paper focuses on “Internet of Things for Estimating Personal Pollution Exposure” and the topic is interesting. The paper needs editing in terms of English and scientific writing style and some parts of the paper are misleading or have no logical order or flow. The paper must go through editing and proofreading process before publication and strictly follow the Guideline for the Author(s). The manuscript has been very casually and carelessly prepared.

The ABSTRACT needs to be revised and special attention should be taken in Language. Moreover, it should be written in proper flow by following; basic introduction, problem, the methodology adopted and results.

Editing and proofreading has been done. The paper follows a logical order which is congruent to the suggested contents in the Guideline for Authors (Introduction, Literature Review, Methodology, Results and Discussion and Conclusions).

The abstract has been rewritten following the suggested structure in the journal template: (1) background (lines 12-13), (2) methods (lines 13-15), (3) results (lines 16-18), and (4) conclusions (lines 18-21) .  

One of the limitations of the paper is that it lacks critical related work but it can be improved before publication. The proposed study is not critically evaluated and compared to the related work/state-of-the-art and is not identified and discussed its drawbacks and limitations. Thus, it is not easy to assess the real contribution of the paper in the field and how much is efficient the proposed solution compared to related works. A clear assessment of the contribution of the authors when compared to existing approaches should be given.

At the end of the literature review, you should have a summary of what the cited literature implies. This summary can be a paragraph to bring together all the pieces of literature you reviewed and to clarify where your research stands. In addition, I feel that additional and more recent works are needed to be examined.

Line 131 (section PAPE Measurement Techniques) critically assesses the current state-of-the-art and in fact, categorizes the existing approaches in the last decade into 3 different groups where the transition from the traditional method to the current state-of-the-art was discussed in detail (line 138). The limitations and drawbacks have also been identified and discussed (line 145-149, 159-160).

We have added more references as marked in green text in the updated update manuscript. Also, we summarized the references to deeply discuss and assess previous studies, and the contribution of the paper. For instance, we included more references for indoor environment monitoring and discuss about the IoT sensor network in the air pollution monitoring perspectives.

Line 193 (Opportunities and Challenges section) for instance, summarizes the drawbacks/gaps in the current methods as discussed in the literature review and discusses how our paper addresses such gaps.

The reported results are interesting but they can be polished by adding more explanation.

Specific discussion has been introduced in section 4, in order to bring more explanations and potential impacts of the claimed contribution.

The Development Issues are not discussed so it needs to be better described.

For the development part, we provide a supplementary material for explaining the development issue It consists of the experimental design, materials and methods used and other development method in this study.

A better discussion and analysis should be made. What exactly it denotes? Why it is important?

More details regarding the implementation, including adopted standards have been included.

The reference and citation formats are wrong and most importantly most of the references and figures are not cited with proper captions.

We have changed the citation and references format in accordance to the ACS style as the journal requests, and adjusted the figures and tables with proper captions as suggested (line 227, 277, 319, 325, etc). Moreover, all figures and tables have been cited in the main text as Figure 1, Table 1, etc. as seen from lines 213 and 318 for instance.

Overall, the manuscript in the current form is not up to the standard as per the novelty and standards of the MDPI journal. So, in my view, the manuscript should be seriously rewritten in both research design and writing style

Reviewer 2 Report

This manuscript reports a framework of internet of things for estimating personal pollution exposure to both indoor and outdoor pollutants. Personal pollution exposure is an important topic that is the focus of indoor air science. The use of internet of things to establish the predicting framework is interesting and promising in dealing with personal pollution exposure. In general, this manuscript is well written and easy to read. This reviewer has the following two comments that may be considered when revising the manuscript.

The first is that the objective of this study is now not well deduced through literature review. In addition, the importance and significance of the work should be stated clearly in the context of both internet of things and personal pollution exposure.

The second is that the organization of this manuscript needs improvement. Now there are many subtitles. Their hierarchical structure should be more clear.

Author Response

This manuscript reports a framework of internet of things for estimating personal pollution exposure to both indoor and outdoor pollutants. Personal pollution exposure is an important topic that is the focus of indoor air science. The use of internet of things to establish the predicting framework is interesting and promising in dealing with personal pollution exposure. In general, this manuscript is well written and easy to read. This reviewer has the following two comments that may be considered when revising the manuscript.

Thank you for your positive comments.

The first is that the objective of this study is now not well deduced through literature review. In addition, the importance and significance of the work should be stated clearly in the context of both internet of things and personal pollution exposure.

We again did a thoroughly literature review (The change in the green color). Moreover, we summarized the references to deeply discuss and assess previous studies, and emphasize the contribution of the paper from IoT, sensor network and APE perspectives.

The second is that the organization of this manuscript needs improvement. Now there are many subtitles. Their hierarchical structure should be more clear.

We followed a logical order of structure which is congruent to the suggested contents in the Guideline for Authors (Introduction, Literature Review, Methodology, Results and Discussion and Conclusions). Section 2.4 Opportunities and Challenges in the literature review structure for example, was necessary to summarize the gaps in the literature review and discuss how our paper contributes to these gaps. In the Methodology section 3, the hierarchy was mainly based on the major contributions of our paper which are (1) the framework (hence section 3.1 Framework) and the (2) IoT application (hence section 3.2 Madrid Case Study. The subsections 3.1.1 and 3.1.2 are necessary to highlight our proposed system architecture and PAPE measurement approach. The subsections 3.2.1-3.2.8 are necessary to discuss in a clearer manner the different parts of our methodology. For instance, we separated the discussion among outdoor pollution modelling, indoor pollution modelling, and their integration, to highlight how we were able to integrate data coming from two different environments. And also to highlight that there are different approaches to measure PAPE based on the type of environment (i.e. indoor/outdoor).

Reviewer 3 Report

In the paper "Internet of Things for Estimating Personal Pollution
Exposure" the authors present a framework for personal APE measuring.
It is an exciting solution. From a technical point of view, this solution
overcomes many problems concerning data collection and pollution measurements.

However, there are two critical issues:

- Meteorological background. The air pollution dispersion depends on local conditions. That is very prominent over an urbanized area. The city ventilation's channels, roughness, tunnel effect in street canyons should be taken into account in the case of personal APE. Moreover, the state of the atmospheric boundary layer (notably, the presence of thermal inversions) can modify spatial distribution and migration of pollutant at person's level (ca. 2m a.g.l.).

- Interpolation method. The authors used simple interpolation methods as IDW and simple kriging. These methods will work very well over flat and open areas with uniform roughness.
Because the authors validated by cross-validation - it looks like "it worked". The results will be significantly worse in street canyons and a city placed on complex terrain. If focusing on geostatistics, a co-kriging method could provide better results. However, The WRF model with a good and precise representation of terrain (DEM, landcover) will be more helpful.  The volunteer was equipped with a sensor. What were the differences between measured and modeled values?

Few questions to answer in the text:
What was the case study period?
What was the synoptic situation during the case study?
How does the study area look like (size, location, denivelation, terrain complexity, type of buildings and its density)?

The authors should focus on the efficiency and feasibility of the system (as they did it), but they cannot neglect fundamental problems related to pollutant dispersion. To make the framework valuable to all stakeholders, the authors should make the solution robust and not only feasible.

Small advice:
The SOS OGC service standard could be helpful to integrate the data streams.

Moreover, the text needs technical corrections.

I suggest a major revision.

Author Response

In the paper "Internet of Things for Estimating Personal Pollution
Exposure" the authors present a framework for personal APE measuring.
It is an exciting solution. From a technical point of view, this solution 
overcomes many problems concerning data collection and pollution measurements.

However, there are two critical issues:

- Meteorological background. The air pollution dispersion depends on local conditions. That is very prominent over an urbanized area. The city ventilation's channels, roughness, tunnel effect in street canyons should be taken into account in the case of personal APE. Moreover, the state of the atmospheric boundary layer (notably, the presence of thermal inversions) can modify spatial distribution and migration of pollutant at person's level (ca. 2m a.g.l.). 

- Interpolation method. The authors used simple interpolation methods as IDW and simple kriging. These methods will work very well over flat and open areas with uniform roughness. 
Because the authors validated by cross-validation - it looks like "it worked". The results will be significantly worse in street canyons and a city placed on complex terrain. If focusing on geostatistics, a co-kriging method could provide better results. However, The WRF model with a good and precise representation of terrain (DEM, landcover) will be more helpful.  The volunteer was equipped with a sensor. What were the differences between measured and modeled values?

First of all, many thanks for your comments.

In this research, we gravitated our target on providing a framework for developing a system that can integrating all the function as we proposed in the paper, instead of deeply discuss environmental modelling.

For sure, the state-of-the-art urban modelling, such as CMAQ, have being widely used with high accuracy. Actually, two of the authors have published different machine learning based models dealing with more sophisticated techniques than the ones used in this application.

Considering that the service must operate in real time for a high number of potential users, and considering that those techniques would require heavy computational effort for deploying these models, we preferred to apply the statistical interpolation approach such as Kriging. However, your suggestion is fully meaningful, and we have included a paragraph in the literature review in the paper (lines 137-145 ) to clarify the choice.

Additionally, in section 4 some paragraphs have been introduced to better explain alternatives, including Machine learning based Models. We hope that with this broader perspective helps the practitioners to better understand the adopted approach.

Few questions to answer in the text:
What was the case study period?
What was the synoptic situation during the case study?
How does the study area look like (size, location, denivelation, terrain complexity, type of buildings and its density)?

Alll the questions have been answered into the text and in our supplementary materials.

1)     We have added the details of study areas information into the text part 3.1.1 Madrid Case study.

2)     We have provided a supplementary material to provide more details about the data acquisition tools, data collection, study period, and environmental modelling method.

The authors should focus on the efficiency and feasibility of the system (as they did it), but they cannot neglect fundamental problems related to pollutant dispersion. To make the framework valuable to all stakeholders, the authors should make the solution robust and not only feasible.

Thank you for your recommendation. Here we deeply discuss about the current advance urban modelling in our literature review and future studies in order to provide more valuable information to the relevant stakeholders, and take it into our consideration for the next research plan.

Small advice:
The SOS OGC service standard could be helpful to integrate the data streams.

This idea was adopted as it is very valuable in terms of technical implementation. In addition, it was translated into the body text as well. (line 237-240)

Moreover, the text needs technical corrections.

We have asked for the professional English proofreading to help us correct the text.

I suggest a major revision.

Round 2

Reviewer 1 Report

The manuscript has been improved and most of the raised questions have been addressed, However, the manuscript still requires a round of Language proofread and sentence structures. Finally, the author is advised to use the MDPI numbered format for reference and citations.

Reviewer 3 Report

The paper "Internet of Things for Estimating Personal Pollution Exposure" was reviewed with the conclusion "Major revision."

That was because of two critical issues:

"- Meteorological background. The air pollution dispersion depends on local conditions. That is very prominent over an urbanized area. The city ventilation's channels, roughness, tunnel effect in street canyons should be taken into account in the case of personal APE. Moreover, the state of the atmospheric boundary layer (notably, the presence of thermal inversions) can modify spatial distribution and migration of pollutant at person's level (ca. 2m a.g.l.).

- Interpolation method. The authors used simple interpolation methods as IDW and simple kriging. These methods will work very well over flat and open areas with uniform roughness.
Because the authors validated by cross-validation - it looks like "it worked". The results will be significantly worse in street canyons and a city placed on complex terrain."

The authors did not improve the paper in any significant way.
I suggest rejecting the paper.